# Separating Malicious from Benign Software Using Deep Learning Algorithm

Ömer Aslan 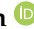

Department of Software Engineering, Bandırma Onyedi Eylül University, Balıkesir 10200, Turkey; oaslan@bandirma.edu.tr

**Abstract:** The increased usage of the Internet raises cyber security attacks in digital environments. One of the largest threats that initiate cyber attacks is malicious software known as malware. Automatic creation of malware as well as obfuscation and packing techniques make the malicious detection processes a very challenging task. The obfuscation techniques allow malware variants to bypass most of the leading literature malware detection methods. In this paper, a more effective malware detection system is proposed. The goal of the study is to detect traditional as well as new and complex malware variants. The proposed approach consists of three modules. Initially, the malware samples are collected and analyzed by using dynamic malware analysis tools, and execution traces are collected. Then, the collected system calls are used to create malware behaviors as well as features. Finally, a proposed deep learning methodology is used to effectively separate malware from benign samples. The deep learning methodology consists of one input layer, three hidden layers, and an output layer. In hidden layers, 500, 64, and 32 fully connected neurons are used in the first, second, and third hidden layers, respectively. To keep the model simple as well as obtain optimal solutions, we have selected three hidden layers in which neurons are decreasing in the following subsequent layers. To increase the model performance and use more important features, various activation functions are used. The test results show that the proposed system can effectively detect the malware with more than 99% DR, f-measure, and 99.80 accuracy, which is substantially high when compared with other methods. The proposed system can recognize new malware variants that could not be detected with signature, heuristic, and some behavior-based detection techniques. Further, the proposed system has performed better than the well-known methods that are mentioned in the literature based on the DR, precision, recall, f-measure, and accuracy metrics.

**Keywords:** cyber security; malware; malware detection; malware dataset generation; deep learning; traditional learning

## 1. Introduction

Internet access among users has been increasing worldwide. These days, more than 50% of the world's population uses the Internet for personal as well as business purposes. This is because digital environments provide tremendous benefits when compared to traditional life without using the Internet. On the other hand, the increased usage of the Internet also raises issues in the cyber security perspective. Cybercriminals spend a vast amount of time obtaining financial and other benefits from companies [1] as well as personal users. According to recent studies, hacking, phishing, spoofing, spyware, and ransomware attacks [2] have risen tremendously. Most of the time, cybercriminals utilize malicious software known as malware to exploit computer system vulnerabilities to launch a cyber attack [3]. Malware is a subclass of software that is intended to perform unwanted actions, such as stealing sensitive data, causing a denial of service attack (DoS), and damage to victim machines [4]. There are different types of malware, such as virus, worm, trojan, spyware, rootkit, and ransomware.

Cyber attacks, which were previously simple and aimless, have been replaced by wider and more targeted attacks. These days, the cybercrime industry has become one of the largest economies in the world. The cost of cyber attacks to the world economy is mentioned in trillions of dollars in many scientific studies. According to Morgan [5], cybercrime is expected to cost nearly USD 10 trillion to the world economy by 2025, and it is expected to increase more in the following decades. The effectiveness of malware as an attack vector in the cyber security domain is increasing rapidly. Recent malware concerns and trends can be viewed in Table 1. The malware increasing rate is exponential, and, currently, there are more than a billion malware samples in the wild. Some of these malware samples are easily accessible on online platforms. Every minute, a few companies across the world become a victim of malware, and there is no antivirus software that can effectively detect the new malware variants. Mobile malware variants are increasing, and IoT devices, cloud environments, banks, and healthcare systems mostly are becoming the target of the malware [6,7].

**Table 1.** Recent malware concerns and trends.

| Parameter Class | Facts on Malware Concerns and Trends |
| --- | --- |
| Economic damage | Cybercrime expected to cost nearly USD 10 trillion to the world economy by 2025 |
| Propagation (speed) | The current number of malware variants is more than 1 billion<br>The malware increasing rate is exponential |
| Spread method/environment | Most of the malware is delivered by email and drive-by download<br>Different malware samples are being sold on DarkWeb |
| Malware target | Mostly small businesses become the target of the malware<br>IoT devices, cloud environments, banks, and healthcare systems mostly become the target of malware |
| Common malware | Trojans are the most common malware types<br>Mobile malware variants are increasing<br>Every minute, a few companies become a victim of ransomware |
| Detectors' efficiency | Antivirus programs do not effectively detect the new malware variants<br>Not resistant to obfuscation techniques |

To protect the computer-based system as well as the communication network, the malware variants must be detected efficiently. There are different malware detection techniques, such as signature, heuristic, behavior, and model checking. The technique names also vary based on the learning method that was used and what the platform was built for, including cloud-based, IoT-based, machine learning (ML)-based, and deep learning (DL)-based. Before the detection phase starts, malware features need to be extracted by using static or dynamic analysis tools. Then, the most significant features are selected for classification. Finally, machine learning algorithms as well as deep learning methods can be used to separate malware from benign files.

A decade ago, signature-based detectors were popular to recognize malware. However, due to polymorphism and packing techniques that new malware variants are using, the signature-based detectors generally fail to detect zero-day malware [8]. The same inefficiency takes place for heuristic techniques as well. Even though the behavior- and model-checking-based detectors can recognize some portion of the unknown malware variants, they still fail to detect more complex malware variants that are substantially different from the existing ones. The machine learning classifiers as well as deep learning methods increase the performance of malware detection techniques.

Machine learning algorithms can be used in many aspects of malware detection [9,10], including feature selection, dimensionality reduction, and classification phases. In these stages, the need for manual work is decreasing while the efficiency of machines is more

involved. In deep learning, even the feature extraction phase can be automated. In this case, the need for domain experts is greatly reduced while the performance of the detectors remains the same or better. Deep learning can be used for various purposes in learning processes, including feature extraction, classification, and dimensionality reduction. Further, it can be combined with other ML models to enhance performance. We assume that, even with domain expert knowledge, the DL model may perform better in some cases. Because of these benefits, we used a deep learning method to separate malicious software from benign samples. Deep learning utilizes several hidden layers instead of one hidden layer, which is used in shallow neural networks. Recently, there are various deep learning architectures proposed to improve the model performance, such as CNN (convolutional neural network), DBN (deep belief network), DNN (deep neural network), and RNN (recurrent neural network) [11–13].

In this study, we aim to detect traditional and more complex malware variants efficiently. To do so, the deep-learning-based malware detection model is proposed. In the proposed model, first, the malware samples are analyzed by using dynamic malware analysis tools, and execution traces are collected. Then, the collected system calls are used to create malware behaviors as well as features. Finally, a proposed deep learning methodology is used to effectively identify malware as well as cleanware. The deep learning methodology consists of one input layer, three hidden layers, and one output layer. In hidden layers, dense (fully connected) layers, which consist of 500, 64, and 32 neurons, are used in the first, second, and third hidden layers, respectively. To increase the model performance and use more important features, various activation functions in the order of Sigmoid, ReLU, Sigmoid, and Softmax are used. The proposed system could effectively distinguish the various malware types from benign files. Further, the proposed system can detect new malware samples that could not be detected with signature- and heuristic-based detection techniques.

In summary, the paper makes the following contributions to build a more robust malware detection model using deep learning:

- The new dataset creation method is proposed;
- The deep-learning-based methodology is suggested to separate malware from benign files;
- Machine learning classifiers are used to detect malware as well;
- The well-known literature studies are reviewed based on the main idea and important findings;
- The proposed method reduces the number of features while increasing the performance significantly;
- The suggested model can effectively detect both known and zero-day malware.

The rest of the paper is organized as follows. Section 2 explains the important points in state-of-the-art studies on malware detection approaches and deep learning malware detection methods. Section 3 presents materials and methods. This section also explains the dataset creation method and the proposed deep learning methodology in detail. Experimental results and discussions are interpreted in Section 4, and limitations and future works are provided in Section 5. Finally, Section 6 presents the conclusion.

## 2. Related Work

This section presents the vast amount of literature studies on malware detection and classification approaches. Based on the method and technology that are utilized, malware detection and classification approaches can be categorized differently. For instance, based on the analysis methods, they can be categorized as static, dynamic, and hybrid malware detection. Based upon the feature selection methods, they can be grouped as signature-, behavioral-, rule-, and model-checking-based detection approaches. Based on the technologies that are used, it can be cloud-based, IoT-based, blockchain-based, machine-learning-based, or deep-learning-based. Malicious software detection is a long process, and several different entities, including technologies, methods, and techniques, are used in

this stage. Malware classification can be defined as one step further to specify the types or families of malicious software after the detection process takes place. In this section, we first would like to categorize the malware detection and classification approaches based on signature-, heuristic-, behavioral-, machine-learning- and deep-learning-based. Then, we review the literature studies in each category by highlighting the main idea and important findings.

### 2.1. Signature-Based Detection

A signature is a sequence of bits that can uniquely identify the program structure. The program signatures are unique; thus, they can frequently be used in malware detection as well as classification. Signature- and heuristic-based malware detection approaches are proposed generally to detect variants of known malware families. However, they cannot effectively detect zero-day malware. Griffin et al. explained an automatic signature extraction technique [14]. The suggested technique could automatically generate the string signatures by using a range of library detection techniques and diversity-based heuristics. According to the authors, the extracted signatures were generally observed in malware samples that were similar to one another. The false positive rate was reduced by using the occurrence probability of arbitrary byte sequences in benign samples.

A signature-based detection method based upon API call tracing was presented by Savenko et al. [15]. The proposed method consists of two parts: the frequency of API calls and the interaction of critical API calls. The malware signature for each program sample was generated from the API calls as well as the interaction of critical API calls. When specifying the program sample containing malicious codes, the signature of the sample is extracted as before and compared with a list of signatures for each malware class. If it matches, it is malware and the class of malware is identified. Otherwise, the analyzed sample is benign. The paper presented that the proposed signature-based method could detect malware class with 92.7% (Delf), 93.1% (Gammima), 93.8% (Bifrose), 96.4% (Ramnit), and 96.56% (MyDoom) accuracy rates.

Sahoo et al. proposed a signature-based detection method for unstructured data in distributed file system Hadoop [16]. They utilized a Clam-AV signature database and used a fast string search algorithm based upon the map-reduce technique. For string matching, Boyer–Moore, Karp–Rabin, and Knuth–Morris–Pratt (KMP) algorithms were used. The proposed method was tested on a real-world dataset. The different accuracy rates were obtained when different pattern-matching algorithms were used. The presented method could not be clearly explained in the paper. Moreover, there was no information about the tested dataset, which detection and accuracy rates were obtained, and how they handle new malware variants.

### 2.2. Heuristic-Based Detection

The heuristic-based detection approach uses experience that utilizes certain rules and ML techniques to separate malware from cleanware. It is effective to detect metamorphic, polymorphic, and some of the previously unknown malware, but it cannot detect complex malware. In the heuristic approach, API system calls, operational code (Opcode), control flow graph (CFG), and hybrid features were used extensively [10,17].

An intelligent malware detection method, which used a heuristic technique, was presented by Ye et al. [18]. The proposed method's goal was to detect previously unseen malware variants and polymorphic malware samples that could not be detected by antivirus scanners. Initially, API sequences of a given program were extracted and appropriate rules were generated using the FP-growth algorithm. Then, classification algorithms were used to detect malware as well as benign. According to the paper, even though the suggested method's performance was better than some antivirus scanners to detect malware variants, it did show the same performance when detecting unknown malware.

Bilar explained Opcode frequency distributions to detect metamorphic as well as polymorphic malware variants [19]. Further, 67 malware samples were disassembled,

and their statistical opcode frequency distributions were compared with the statistics of 20 benign samples. The experiment test results indicated that there was a significant difference in the opcode distribution of malware and benign samples. The presented method was tested on a small portion of malware and benign samples; more samples need to be analyzed to obtain more reliable results.

### 2.3. Behavioral-Based Detection

The behavioral-based approach monitors the sample program behaviors, and, based on the behaviors that are obtained, the sample program is marked as malicious or benign. Behavioral-based detection approach has seemed to be a promising solution to detect both known and unknown malware variants for a decade. However, some malware variants do not display their actual behaviors in environments such as virtual machines and sandboxes and, hence, cannot be detected by behavioral-based detection approaches.

A system-centric behavioral model, which separated malware from benign samples, was presented by Lanzi et al. [20]. The paper presented that the way malicious samples interact with system resources, including directories, registries, etc., was different from the interaction of benign samples. Behavior sequences were extracted from the system calls by using the interaction differences. Finally, malware and benign categories were identified by using behavior sequences. The n-gram technique was used when the behavior sequences were generated. Thus, the proposed method was not very effective to separate malware from benign samples because of the excessive number of sequences of behaviors generated with the n-gram.

Galal et al. proposed a behavior-based malware detection model that relies on malicious actions displayed by malware as well as benign samples [21]. To extract the malware features, dynamic analysis was performed on collected malware samples in a controlled virtual environment, and API call traces were captured. Then, the API call traces were converted into high-level features. Finally, machine learning classifiers were used, including decision tree (DT), random forest (RF), and support vector machine (SVM), to detect malware. According to the paper, a high accuracy rate was obtained in the detection of malware variants. However, the numerical statistics, such as detection rate, accuracy rate, etc., were not provided in the paper.

Ding et al. presented a malware detection technique based upon a family dependency graph [22]. First, system calls were obtained from the malware and benign samples. Then, a dynamic taint analysis method was used to identify the dependency relationship among the system calls. After that, family dependency graphs were generated. Based upon the generated dependency graphs, common behavior graphs were extracted to represent the malware family's behavioral features. In the end, graph-matching, which used maximum weight subgraphs, was used to recognize malware samples. According to the paper, the suggested method could detect different malware samples with a high detection rate. However, for some malware variants, the detection rate was low, and false alarm rates were high.

### 2.4. Machine-Learning-Based Detection

Machine-learning-based malware detection methods have become popular after 2015 and still are used in many scientific studies. Malware detection, which used machine learning, was proposed by Markel [23]. The metadata, specifically header data from each Windows Portable Executable (PE32), were collected. Then, learning methods were performed on collected metadata to detect malware. The presented method was tested on realistic datasets. Based on the test results, it was found that classifiers did not return satisfactory results on test data with a low malware prevalence. Even with ensemble learning, satisfactory results were not obtained. It was found that, to obtain the best performances, the training and testing data should be in the same proportion.

Sethi et al. proposed a machine learning detection as well as classification schema [24]. The Cuckoo sandbox was used to collect the system activities when program samples

were executed. Then, on collected activities, a property extraction, as well as a selection method, were proposed and performed to separate malware from benign samples. Finally, various machine learning algorithms were performed to detect malware as well as fine-grained classification. According to the paper, the obtained detection and classification results were high when compared with the state-of-the-art methods. However, the obtained performance results for some machine learning classifiers were low. For instance, for SVM and RF, the accuracy rate was obtained as 86.7% and 88.2%, respectively, which was much lower than the deep learning methods.

Singh examined the malware detection techniques that used machine learning algorithms [25]. The paper discussed several challenges in detecting malware when a traditional approach was used. When machine learning techniques were applied to the malware features, the malicious patterns were recognized effectively. Additionally, machine learning assists to detect unknown variations while reducing time and human intervention during the detection processes. In addition, machine learning techniques were commonly used in signature- and behavior-based detection approaches to increase the detection rate as well as accuracy.

*2.5. Deep-Learning-Based Detection*

Recently, deep-learning-based malware detection and classification methods have appeared to be improving the accuracy and resiliency of the detectors. In deep learning models, there are generally two main techniques to extract malware features before classification takes place. In the first technique, malicious binary files are converted into images and then features are extracted. In the second one, execution traces of malware are collected by using relevant malware analysis tools, and then execution traces are used to visualize malicious files. After the feature extraction phase is complete, various deep learning architectures can be used to classify the malware. In the last two years, there has been growing interest in malware detection approaches that use various deep learning models [26–32].

Kumar proposed an MCFT-CNN (fine-tuned convolution neural networks) method to classify malware [26]. The method utilized deep transfer learning by enhancing the ResNet50 model to categorize different malware families. The suggested model was tested on Malimg and Microsoft BIG 2015 datasets. The accuracy was measured as 98.63% for the Microsoft dataset. In [27], the authors used LSTMs (long short-term memory networks) to detect malware on Windows audit logs. Initially, they extracted properties from Windows audit logs and then used one-hot encoding to transform them into continuous values. After that, they used LSTMs to recognize malware from benign files. Even though the proposed method achieved satisfactory DR, several false alarms reduced the efficiency of the model. To improve the model performance, the false alarm rate must be decreased, and a larger dataset is needed to test the model.

Jian et al. suggested a new malicious software detection framework using deep neural networks [28]. The benign and malware samples were collected and visualized by using an IDAPro disassembler. To obtain the high-dimensional features, visualized bytes and asm files were converted into three-channel RGB images. Finally, a deep neural network model using SEResNet50 + Bi-LSTM was used to detect malware and classify the malware families. The proposed framework was tested on Microsoft BIG 2015 dataset, and a 98.3% accuracy rate was obtained. To validate the presented method more clearly, the method will need to be tested on different datasets.

Baek et al. suggested a hybrid malware detection model, which consists of two phases, to detect malware on IoT devices [29]. In the first phase, opcode was generated by performing static malware analysis, and benign files were identified by using Bi-LSTM (bidirectional long short-term memory). In the second phase, dynamic analysis was used to extract malware features, and then malware was detected by using the EfficientNet-B3 model. The accuracy rate of static and dynamic analysis was measured as 94.46% and 94.98%, respectively. The presented method was not compared with other state-of-

the-art methods enough and the contents of the analyzed malware samples were not explained enough.

In our previous study [30], we presented a hybrid deep learning architecture to categorize the malware families. In the proposed model, first, the malware samples were converted into grayscale images. Then, the high-level malware features were generated by using pre-trained networks via ResNet-50 and AlexNet. Finally, the generated features were classified by using a deep neural network as a supervised learner to classify malware families. The presented method was evaluated on Microsoft BIG 2015, Malimg, and Malevis datasets. The accuracy rate on the Malimg dataset measured 97.78%, which was substantially high when compared with ML-based detectors.

A convolutional-neural-network-based deep learning framework was described in [31]. The proposed framework used transfer-learning-based architecture, which utilized spatial convolutional attention to classify 25 malware families. The framework performances were tested on the Malimg dataset and obtained precision, recall, specificity, F1, and accuracy measured as 97.1%, 96.9%, 97.3%, 97%, and 97.4%, respectively. The presented method was tested on the Malimg dataset; it can be tested on more malware datasets.

Azeez et al. examined ensemble-learning-based models to detect malicious software [32]. In the study for the base classification stage, a stacked ensemble of fully connected and 1D convolutional neural networks (CNNs) was used. For the final stage of classification, machine learning algorithms were used. For meta-learner, 15 ML algorithms were used, and, for comparisons, AdaBoosting, decision tree, gradient boosting, naïve Bayes, and random forest were used. The method was tested on malware and benign samples, which were collected from the Kaggle dataset. Among 14.599 malware samples, the best accuracy result was measured as 98.62% when XGBoost was used. In some cases, the accuracy rate was rather low. The presented schema can be tested on different datasets as well as unknown malware variants.

*2.6. Evaluation of Related Works on Malware Detection Approaches*

In this section, several state-of-the-art studies were summarized in different categories, including traditional signature-, heuristic-, behavioral-, machine-learning-, and deep-learning-based (Table 2). The signature- and heuristic-based malware detectors are fast and efficient to detect traditional malware, but they fail to detect zero-day malware. Behavioral-based and machine-learning-based malware detectors are efficient to recognize some portion of the zero-day malware, but they fail to detect malware variants whose behavioral patterns are relatively different from the existing ones. DL-based detectors generally visualize the malware executable and decrease the need for feature creation as well as the selection process most of the time. This case increases the detection rate for both known and unknown malware. However, there are still some problems related to machine learning and deep learning when detecting and classifying malware. These problems can be listed as follows:

- ML and DL algorithms make assumptions about the data and these assumptions can be wrong sometimes;
- Sometimes, there is not enough information within flows, need contextual features;
- Generally, data contain high dimensionality, which may decrease system performance;
- The algorithms do not take domain expert knowledge into account; this can generate meaningless features;
- ML algorithms are prone to bias and cannot handle outliers all the time;
- Detecting zero-day malware is challenging;
- The intelligent malware evades the ML- and DL-based classifiers.

Most of the malware classification approaches, which use deep learning models, were tested on Microsoft BIG 2015, Malimg, and Malevis datasets. There are a few deficiencies in these datasets. For instance, most of the malware in the datasets is already known malware; it is not up to date. There are no benign samples in some of these datasets, and there are not enough balanced samples for each malware family. Most of the studies that

used these three malware datasets were performing classification tasks that categorized some malware families; they were not recognizing malware from the benign class. Further, the analyzed DL-based malware classification methods are similar in most of the studies, and these studies could not make a major difference in effectively recognizing the new malware variants. The proposed method tried to eliminate the leading current DL-based classification state-of-the-art studies and made necessary contributions in this area side by side to effectively recognize both known and new unknown malware.

**Table 2.** The summary of the state-of-the-art studies on malware detection and classification methods.

| Scientific Research | Year | Feature Extraction Method/Feature Representation | Detection Method | Success |
|---|---|---|---|---|
| Griffin et al. [14] | 2009 | Range of library detection techniques and diversity-based heuristics | Signature-Based | Not given |
| Sahoo et al. [16] | 2014 | Unstructured data in Hadoop by using a fast string search algorithm | | Not given |
| Savenko et al. [15] | 2019 | Frequency of API calls as well as the interaction of critical API calls | | 92.7% accuracy |
| Ye et al. [18] | 2007 | API sequences with FP-growth algorithm | Heuristic-Based | 93.07% accuracy |
| Bilar [19] | 2007 | Opcode frequency distributions | | Not given |
| Lanzi et al. [20] | 2010 | Behavior sequences extracted from the system calls | Behavior-Based | 91% detection rate |
| Galal et al. [21] | 2016 | High-level features were represented as API calls | | 93.98% accuracy |
| Ding et al. [22] | 2018 | Dependency graph by using taint analysis | | 91.6 accuracy |
| Markel [23] | 2015 | Header metadata from Windows Portable Executable | Machine-Learning-Based | Low performance |
| Sethi et al. [24] | 2019 | Dynamic system activities | | 88.2% accuracy |
| Singh [25] | 2021 | Runtime features | | Depends on the study |
| Kumar [26] | 2021 | Fine-tune convolution neural networks with ResNet50 | Deep-Learning-Based | 98.6% accuracy |
| Baek et al. [29] | 2021 | Bi-LSTM with EfficientNet-B3 model | | 94.9% accuracy |
| Aslan and Yilmaz [30] | 2021 | Grayscale images with ResNet-50 and AlexNet | | 97.7% accuracy |
| Awan et al. [31] | 2021 | Transfer learning based on spatial convolutional attention | | 97.4% accuracy |

## 3. Materials and Methods

This section explains materials and methods. The proposed system architecture consists of three modules, namely data collection as well as labeling, feature creation representation, and classification, which are shown in Figure 1. The main idea is to take the malware and benign binary files as input to the proposed system and produce a result that shows whether the binary contains malicious code or not. To increase detection performance, domain expert knowledge and deep learning models are combined in this study. This is because deep-learning-based models can easily be deceived by evasion attacks in the cybersecurity domain, whereas combining domain knowledge with deep learning increases the resistance level against evasion attacks.

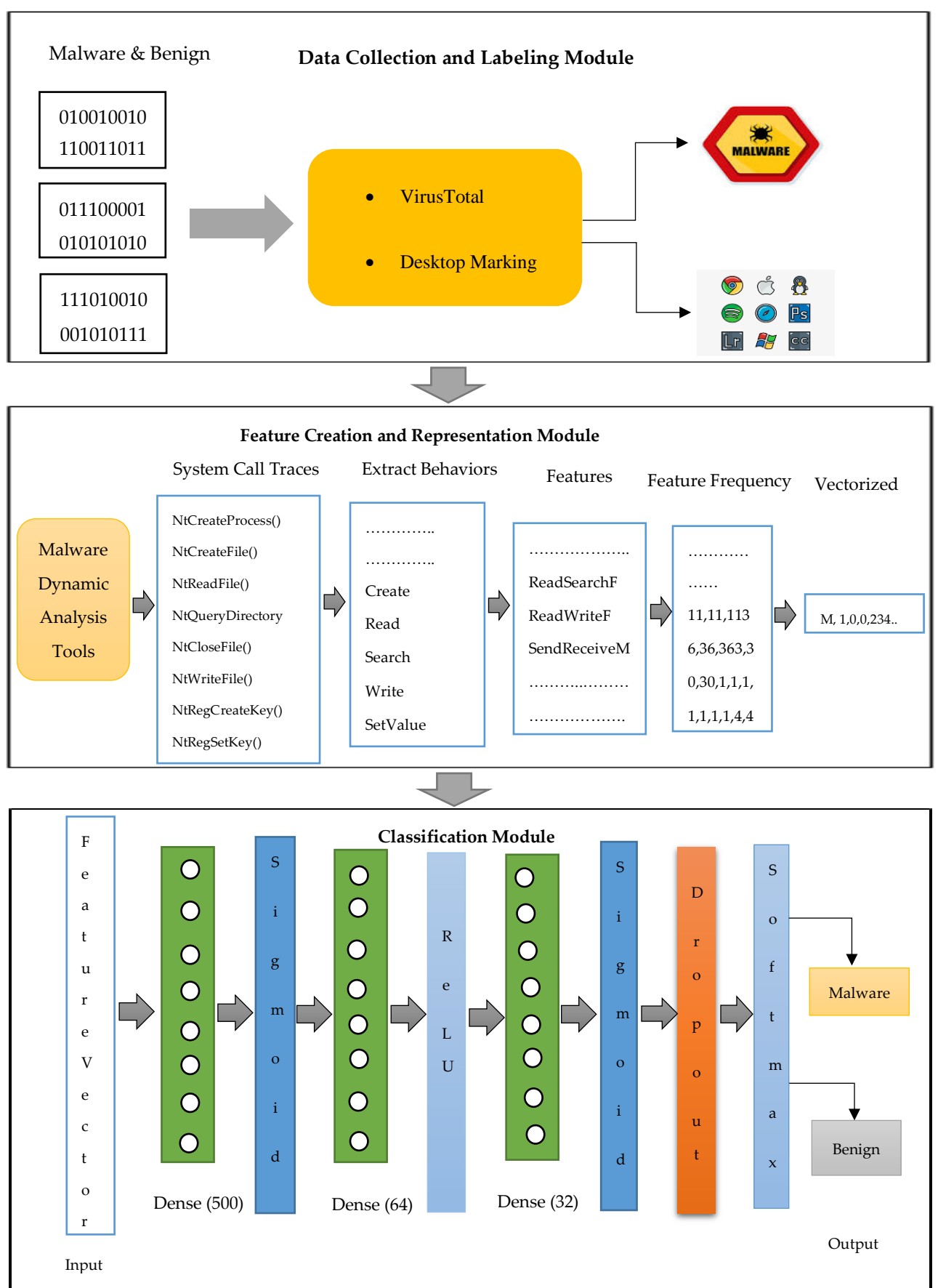

**Figure 1.** Proposed malware detection architecture.

In the first module, various malware and benign samples are collected from different sources and labeled by using VirusTotal and desktop marking (Figure 1). VirusTotal is a website that uses various vendor antivirus scanners to label the given program. For desktop marking, several types of antivirus software are downloaded into the desktop and used. Further, some basic features that are gathered during the malware analysis process are taken into consideration during the labeling stage. In some cases, it was difficult to label the malware samples by using an antivirus scanner; for those cases, extra features are used.

After the labeling stage is completed, we execute each sample in the protected environment in order to specify malware and benign features. For that, first, the malware and benign samples are analyzed under dynamic analysis tools and system calls execution traces are collected. Second, system calls are grouped into six categories, including process, thread, file, registry, memory, and network. Based on the six categories, the behaviors are created. Third, features are generated from the behaviors. In this process, behaviors themselves, in which locations they are executed, and some of the other parameters, including PID (process identifier), parent PID, results, time, event class, and sequence are considered. Finally, feature frequency and the row feature vector are generated. In the feature creation and representation process, the binary files that consist of hundreds of thousands of {0, 1} are taken, and feature vectors are generated that comprise a few hundred natural numbers $f = \{0, 1, 2, 3, \ldots, n\}$, where $f$ is feature set and $n$ is the last number of values. We think that examples of malware and benign files display behaviorally similar sequences in similar frequency.

In the classification module, the new deep learning methodology is presented to detect malware as well as benign samples. The methodology consists of 1 input, 3 hidden, and 1 output layer (Figure 1). In hidden layers, dense (fully connected), which consists of 500, 64, and 32 neurons, are used in the first, second, and third hidden layers, respectively. To keep the model simple as well as obtain optimal solutions, we have selected 3 hidden layers in which neurons are decreasing in the following subsequent layers. To increase the model performance and use more significant features, various activation functions in the order of Sigmoid, ReLU, Sigmoid, and Softmax are used. To decrease the error between the training and testing, the dropout rate of 60% is applied. A detailed explanation of the proposed system can be viewed in the following subsections.

### 3.1. Dataset Creation and Representation Method

Since the malware can easily change its static features to escape from the detection system, the dynamic features are considered in this study. In order to generate dynamic malware and benign features, the real active malware and benign samples are collected and executed under virtual machines. It is known that the way malware samples run on operating system resources is different from benign samples. Based on this fact, the execution traces of system calls are collected for each sample. During the execution trace collection, the processes that are created by each sample that performed system calls are also considered. The collected system calls are classified based upon the operating system resources, including process, thread, file, registry, memory, and network. Dynamic analysis tools, including Autoruns, Process Monitor, Regshot, and Wireshark, are used to capture the system calls execution traces. For each system call, the available arguments, which provide extra information on how the system call is performed, such as process name, PID (process identifier), TID (thread identifier), parent PID, operation, path, image path, results, time, event class, sequence, category, and detail, are collected as well. These arguments are used during the behavior and feature creating process. For each malware and benign sample, thousands of system calls are obtained that are difficult to handle during the detection process. Due to that, the high-level behaviors are generated based on the performed system calls. We defined behavior as a group of system calls that perform meaningful actions on operating system resources.

The created malware and benign behaviors are classified based on the properties, such as resource types, executed paths, behavior types, PID, parent PID, called DLLs, etc. Then,

the level of behaviors, which represents the probability of being observed in malware and benign samples, is calculated. For instance, writing normal files and system files, such as svchost. exe, wininit. exe, winlogon. Exe, is different, or reading a normal file content and process memory is different. Even if both behaviors are the same, the probability of these behaviors being viewed in malicious and benign samples will be different. The calling of Kernel32.dll or Ntdll.dll is different because benign samples generally call Kernel32.dll while malware calls Ntdll.dll in order to hide its action. Moreover, malware samples more frequently copy themselves into the computer Autostart file and registry location than benign samples. These kinds of behaviors are considered more during the feature creation as well as selection processes. The level of behavior is important during the feature engineering process. It is because, the higher the level of behavior, the more likely this behavior will be selected in the feature set.

The malware and benign features are generated by using the relationship among the behaviors as well as the level of behaviors. When features are generated from the behaviors, top ten consecutive behaviors are taken into consideration. When n-gram is used, only the specified $n$ value is used during the feature generation process. For instance, if the order of the behaviors is $\{a_1, a_2, a_3, b_1, b_2\}$ ($a$ and $b$ show different system resources) and $n$ is 2, the generated features will be $\{<a_1, a_2>, <a_2, a_3>, <a_3, b_1>, <b_1, b_2>\}$. For the same behavior set, if $n$ is 4, the generated features will be $\{<a_1, a_2, a_3, b_1>, <a_2, a_3, b_1, b_2>\}$. When n-grams are used this way, many important features will be overlooked. In our proposed method, we consider the top ten consecutive behaviors each time during the feature generation. For the same behavior set, the generated candidate features will be $\{<a_1, a_2>, <a_1, a_3>, <a_2, a_3>, <a_1, a_2, a_3>, <b_1, b_2>\}$. Based on the system resource instances and relationships, the feature set will be generated from the candidate features. Further, the features whose level values are below a certain threshold value are also not included in the feature set. That way, the generated feature set will contain much fewer features than the candidate feature set.

When $n$ grows, the number of features created using n-grams becomes very large, while the number of features does not increase after a certain number in the dataset created with the proposed method. When the number of behaviors is 50.000 and $n$ is 2, ($n - 1$) features 49.999 will be generated. For the same number of behaviors, when $n$ is 3 and 4, ($n - 2$) and ($n - 3$) features 49.998 and 49.997 will be generated, respectively. For the proposed method, around 100 features are generated for the same behavior set, which is much fewer than n-gram. This is the case because, during the feature generation process, we first group the behaviors based on the operating system resources, including process, thread, file, registry, memory, and network. Then, a candidate feature set is created when the behaviors on the same resource instance are performed. If a feature is not performed in a specific resource path, it is also removed from the dataset. Finally, the frequency of feature values is used when a row feature is created for each sample.

After the feature generation process is completed, the feature frequency is calculated for each feature and written as a feature value. Finally, the row feature vector is created for each program sample using the feature frequency. That way, our proposed feature generation method took malware and benign binary files as input and generated a row feature vector for each sample. The proposed feature creation method data flow stages can be summarized as follows:

- Stage 1: The malware and benign samples are employed under the dynamic malware analysis tools;
- Stage 2: The execution traces of system calls are obtained;
- Stage 3: System calls are converted into behaviors;
- Stage 4: Behaviors are classified based on the resource types and paths;
- Stage 5: The importance level of behaviors is calculated;
- Stage 6: The features are generated based on the relationship among the behaviors as well as the level of behaviors;
- Stage 7: The feature frequency is calculated for each program sample;
- Stage 8: The row feature vector is created for each program sample.

*3.2. Classification Model*

In this subsection, the proposed deep learning methodology as well as used ML algorithms are explained for classification. The subsection is divided into two sections: the proposed DL methodology and the used ML algorithms.

3.2.1. Using the Proposed Deep Learning Methodology to Detect Malware

Deep learning is a subclass of machine learning that was inherited from artificial neural networks. In deep learning, high-level features can be learned through the layers. Deep learning consists of 3 layers: input, hidden, and output layers. The inputs can be in various forms, including text, images, sound, video, or unstructured data. The idea is to extract high-level features with no human intervention or with less domain knowledge. During the malicious software recognition stage, DL can be used for feature extraction, classification, or also can be used for both. There are various DL architectures and models that can be used for different problem domains. The most well-known DL architectures and models can be listed as convolutional neural networks (CNNs), deep belief networks (DBNs), deep reinforcement learning (DRL), long short-term memory networks (LSTMs), and recurrent networks (RRNs). Recently, some of these architectures and models have started to be used in malware detection as well as classification.

The proposed deep learning methodology is performed on our dataset that was created in the previous subsection. First, the preprocessing stage is performed to eliminate missing values and convert the string into numerical values. For the learning phase, 1 input layer, 3 hidden layers, and 1 output layer are used. The used hidden layers are dense (fully connected) layers that consist of 500 neurons in the first hidden layer, 64 neurons in the second hidden layer, and 32 neurons in the third hidden layer. Three different activation functions and one optimization function are used. The dropout is employed to decrease the risk of overfitting. After the training is completed, the malware samples are identified as malicious or normal by the output layer. Given input features $X$ ($x_1$, $x_2$, $x_3$, $\ldots$, $x_n$), weights $W$ ($w_1$, $w_2$, $w_3$, $\ldots$, $w_n$), and bias $b$, the sum of features is calculated as

$$\sum_{x=1}^{n} (x_i w i) + b \tag{1}$$

The sum of the calculated values is input into the activation function to generate the hidden layer's neurons as well as an output layer. Initially, weights are assigned randomly. During the training, weights values are changed based on the Sparse Categorical Cross Entropy loss and Adam optimizer. The used hyperparameters for our deep learning methodology can be viewed in Table 3. To increase the deep network learning capacity, we utilized several activation functions in order of Sigmoid, ReLU, Sigmoid, and Softmax. The activation function transforms the sum of the given input values (output signals from the previous neurons) into a certain range to determine whether it can be taken as an input to the next layer of neurons or not. The Sigmoid, ReLU, and Softmax activation functions are calculated as the following:

$$\text{Sigmoid}: \ f(a) = \frac{1}{1 + e^{-a}} = \frac{e^a}{1 + e^a} \tag{2}$$

$$\text{ReLU}: \ (0, \ a) = 0, \ if \ a < 0; \ (0, \ a) = a, \ if \ a \geq 0 \tag{3}$$

where $a$ is the sum of the previous layer neurons output. To increase the generalization capacity of the proposed deep learning methodology while reducing the overfitting, the dropout 60% is used.

$$\text{Softmax}: \ \sigma(Z)_i = \frac{e^{z_i}}{\sum_{j=1}^{K} e^{z_j}} \tag{4}$$

where $i$ = 1, 2, 3, $\ldots$, $K$ (number of classes) and $Z$ (input vector) = ($z_1$, $z_2$, $z_3$, $\ldots$, $z_k$). We used the Adam optimization algorithm to train our network. In the Adam algorithm, the

estimation of the first- and second-order moments are used to adapt the learning rate for each weight.

**Table 3.** Hyperparameters of used deep learning models.

| Parameter | Value |
|---|---|
| Batch Size | 64 |
| The number of hidden layers | 3 |
| Epochs | 50 |
| Dropout Rate | 0.6 |
| Learning Rate | 0.01 |
| Activation Function | Sigmoid, ReLU, Softmax |
| Loss Function | Sparse Categorical Cross Entropy |
| Optimizer | Adam |

3.2.2. Using Machine Learning Algorithms to Detect Malware

After the feature creation and selection processes were completed in the previous module, the training and testing phases were performed. In this module, machine learning classifiers, including NB, BN, Decision Stump, SMO, AdaBoost, and LogitBoost, were performed to separate malware samples from benign ones. These classifiers can be explained shortly as the following:

**NB** is a probabilistic ML classifier that does not require much computation time and produces good results with high-dimensional data. It may not return good results because it calculates assumptions that are not very related to one another. In NB, given $n$ features $(x_1, x_2, x_3, \ldots, x_n)$ and $m$ classes $(c_1, c_2, c_3, \ldots, c_m)$, conditional probability for class $C_k$ calculated as

$$P(C_k/X) = P(C_k) \, P(X/C_k)/P(X) \tag{5}$$

where $X = (x_1, x_2, x_3, \ldots, x_n)$, and $1 \leq k \leq m$.

$P(X/C_k)$ can be calculated as

$$\prod_{i=1}^{n} P(x_i/C_k) \tag{6}$$

$$P\left(\frac{X}{C_k}\right) = P\left(\frac{x_1}{C_k}\right) P\left(\frac{x_2}{C_k}\right) P\left(\frac{x_3}{C_k}\right) \ldots P\left(\frac{x_n}{C_k}\right)$$

In our case, $k = 2$, malware (1) or benign (0). We can recognize the type of malware when $k$ is larger.

**BN** is a probabilistic ML classifier that uses a graphical model where the vertex corresponds to variables and edges representing conditional probability. It generally returns fast and efficient results. However, it is not very practical for datasets with many features.

**Decision Stump (DS)** is an ML classifier that consists of a one-level decision tree. Each time, the algorithm considers one feature and finds the point that can separate the data most. In other words, only one feature at a time in the decision tree is used for classification. If input value $x$ is larger than the value $a$ ($x > a$), then the class of $x$ is specified as $c_1$; otherwise, the class of $x$ is specified as $c_2$.

**SMO** is an ML optimization algorithm that is used during the training of SVM (support vector machines) when solving the quadratic programming problem. Given the set of input vector $X_i$ and corresponding class label $Y_i$: $(X_1, Y_1), (X_2, Y_2), \ldots, (X_n, Y_n)$; a SVM training by solving a malware problem can be expressed in the dual form as follows:

$$max_\alpha \sum_{i=1}^{n} \alpha_i - \frac{1}{2} \sum_{i=1}^{n} \sum_{j=1}^{n} y_i y_j \, K(x_i, x_j) \alpha_i \alpha_j \tag{7}$$

Subject to $0 \leq \alpha_i \leq C$, $for\ i = 1,\ 2,\ 3,\ \ldots,\ n$

$\sum_{i=1}^{n} y_i \alpha_i = 0$, where $C$ is a hyper parameter in SVM

$K\ (x_i,\ x_j)$ is the kernel function, and $\alpha_i$ values are Lagrange multipliers.

**AdaBoost** is a statistical ML algorithm that is utilized from multiple classifiers to improve the model performance. The goal is to minimize the training error by setting weights of weak learners in each iteration. AdaBoost produces satisfactory results for binary classification.

**LogitBoost** is another boosting ML algorithm that minimizes the logistic loss during the training. In some studies, even higher performance values were obtained by using classical ML classifiers, including NB, BN, DS, SMO, Adaboost, and LogitBoost; in our case, we obtained better performance with deep learning algorithms.

### 3.3. Case Study

This section explains the case study and test environments. Various versions of Windows virtual machines, including Windows 7, 8, and 10, as well as real machines, were used in this study. The collected malware and benign samples were performed on virtual machines. Totally, ten thousand program samples are tested on these virtual machines; 70% of them are malware samples, while 30% are benign ones. The analyzed malware samples are from different categories as well as families (Figure 2). In Figure 2, different colors are used to show the malware types in level as well as various families. For instance, yellow colors represent the different types of malware samples, while blue colors show the subclass of malware types. In this regard, downloader, dropper, injector, and adware can be viewed as a subclass of Trojan, while Keylogger can be viewed as a subclass of spyware. Pink colors symbolize new malware variants or more complex malwares that use encryption or packed techniques, while the black colors represent various malware families. To capture the execution traces that were performed by malware and benign samples, dynamic malware analysis tools were used. For each sample, a clean version of the operating system was used. The collected execution traces were analyzed by our proposed feature creation model, which was implemented in python scripting language. The detection phase of our model, which was using deep learning, was also implemented in python.

### 3.4. Model Performance and Evaluation

To evaluate the proposed model performance, the holdout as well as cross-validation methods were used. To show the model's effectiveness, different k values in cross-validation and several percentage splits in holdout were used. For all cases, the satisfactory performance results were obtained. The metric values were calculated to measure the model performance by using confusion matrix (CM) (Table 4). TP, TN, FP, and FN stand for the number of malware samples correctly identified as malware, the number of benign samples correctly identified as benign, the number of benign samples mistakenly identified as malware, and the number of malware samples mistakenly identified as benign, respectively. These values were used to calculate the precision, recall, f-measure, and accuracy metrics as the following:

$$Precision = (TP)/(TP + FP) \tag{8}$$

$$Recall = DR = (TP)/(TP + FN) \tag{9}$$

$$F\text{-}measure = (2 * precision * recall)/(precision + recall) \tag{10}$$

$$Accuracy = (TP + TN)/(TP + TN + FP + FN) \tag{11}$$

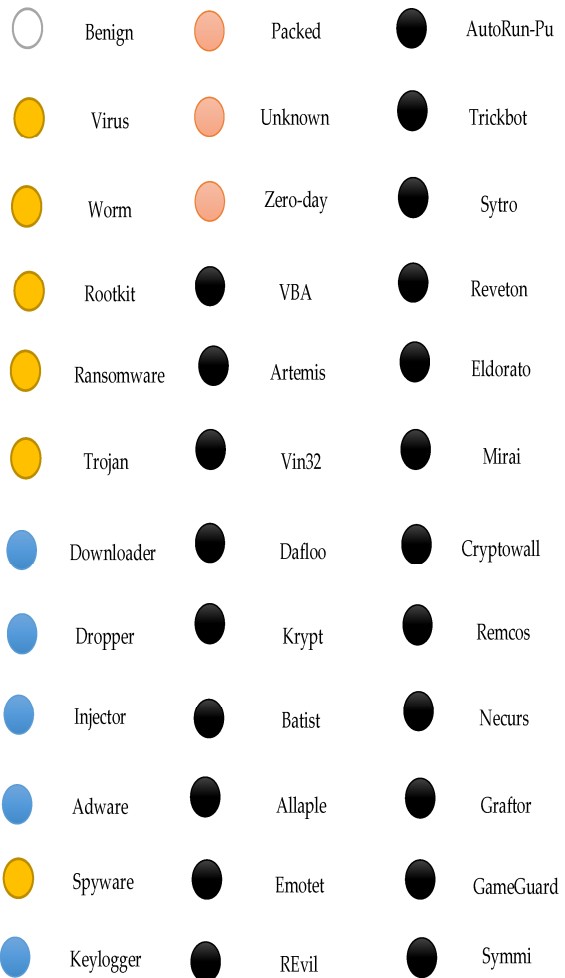

**Figure 2.** Some of the analyzed malware samples (different types, variants, and families).

**Table 4.** Confusion matrix.

| | | Predicted Class | |
|---|---|---|---|
| | | Yes | No |
| Actual class | Yes | TP | FN |
| | No | FP | TN |

## 4. Experimental Results and Discussion

The experiment test results are summarized in Tables 5–8 and in Figures 3 and 4. To evaluate the presented model's performances, various combinations of percentage splits as well as cross-validation were used. Based on the performance metrics that were obtained, the deep learning algorithms performed relatively well on our created malware dataset. Most of the time, the obtained detection rate (DR) and accuracy were measured as 99% and 99.80%, respectively. When we changed the DL hyper parameters, similar performances were obtained as well. This shows that combining the proposed model with an appropriate DL architecture can efficiently separate malware variants from the cleanware. Our model could efficiently separate various malicious software: traditional malware as well as new variants including zero-day malware, from the benign files. When different machine learning algorithms are applied to the same dataset, the obtained performance values are decreased sharply. Even ANN performed poorly when compared with DL models. The proposed model has also been compared with the leading models in the literature.

According to the performance metrics results, the proposed model performed better than well-known state-of-the-art-studies based on the DR and accuracy.

**Table 5.** The performance results when a deep learning model was used.

| Method | No. of Neurons in First Hidden Layer | No. of Neurons in Second Hidden Layer | No. of Neurons in Third Hidden Layer | Benign Malware | Prec (%) | Rec (%) | F1 (%) | Acc (%) |
|---|---|---|---|---|---|---|---|---|
| Deep Learning Using Cross-Validation *k* = 5 | 500 | 64 | 32 | 0 | 99 | 100 | 99.5 | 99.80 |
| | | | | 1 | 100 | 100 | 100 | |
| Deep Learning Using Cross-Validation *k* = 10 | 500 | 64 | 32 | 0 | 99 | 100 | 99.5 | 99.60 |
| | | | | 1 | 100 | 99 | 99.5 | |
| Deep Learning Using Percentage Split (85%, 15%) | 500 | 64 | 32 | 0 | 99 | 100 | 99.5 | 99.73 |
| | | | | 1 | 100 | 100 | 100 | |
| Deep Learning Using Percentage Split (70%, 30%) | 500 | 64 | 32 | 0 | 100 | 99 | 99.5 | 99.67 |
| | | | | 1 | 100 | 100 | 100 | |
| Deep Learning Using Percentage Split (50%, 50%) | 500 | 64 | 32 | 0 | 99 | 99 | 99 | 99.60 |
| | | | | 1 | 100 | 100 | 100 | |

**Table 6.** Performance results when ANN was used.

| Method | Benign Malware | Prec (%) | Rec (%) | F1 (%) | Acc (%) |
|---|---|---|---|---|---|
| ANN Using Percentage Split (85%, 15%) | 0 | 77 | 98 | 86 | 90.8 |
| | 1 | 99 | 88 | 93 | |
| ANN Using Percentage Split (70%, 30%) | 0 | 70 | 99 | 82 | 86 |
| | 1 | 99 | 83 | 90 | |
| ANN Using Percentage Split (50%, 50%) | 0 | 73 | 91 | 81 | 86.7 |
| | 1 | 95 | 85 | 90 | |

**Table 7.** Confusion matrices results when DL versus ANN were used for different number of samples.

| | | Predicted Class | | | |
|---|---|---|---|---|---|
| | | Cross-Validation | | Percentage Split | |
| Actual Class | DL | 445 | 1 | 1517 | 8 |
| | | 3 | 1051 | 12 | 3463 |
| | ANN | 866 | 11 | 1384 | 141 |
| | | 366 | 1757 | 523 | 2952 |

**Table 8.** Summary of machine learning classifiers' results.

| Classifier | Benign Malware | Prec (%) | Rec (%) | F1 (%) | Acc (%) |
|---|---|---|---|---|---|
| NB | 0 | 55.3 | 92.2 | 69.2 | 74 |
| | 1 | 94.8 | 65.7 | 77.6 | |
| BN | 0 | 77.3 | 94.3 | 85 | 89.4 |
| | 1 | 97.1 | 87.2 | 91.9 | |
| Decision Stump | 0 | 86 | 79.3 | 82.5 | 89.4 |
| | 1 | 90.8 | 94.1 | 92.4 | |
| SMO | 0 | 83.9 | 91.5 | 87.6 | 91.8 |
| | 1 | 95.9 | 91.9 | 93.9 | |
| LogitBoost | 0 | 90 | 89.6 | 89.8 | 93.6 |
| | 1 | 95.2 | 95.4 | 95.3 | |
| AdaBoost | 0 | 89.4 | 91.3 | 90.4 | 93.8 |
| | 1 | 96 | 95 | 95.5 | |

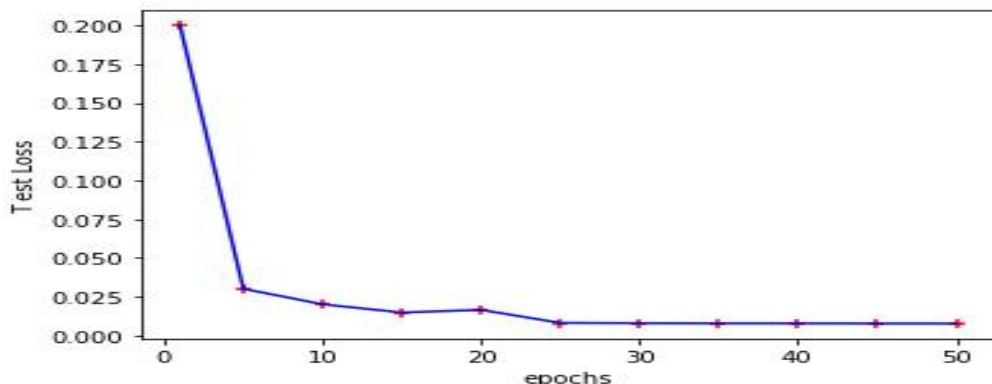

**Figure 3.** Loss (Sparse_categorical_crossentropy) with respect to the number of epochs when detecting malicious executable.

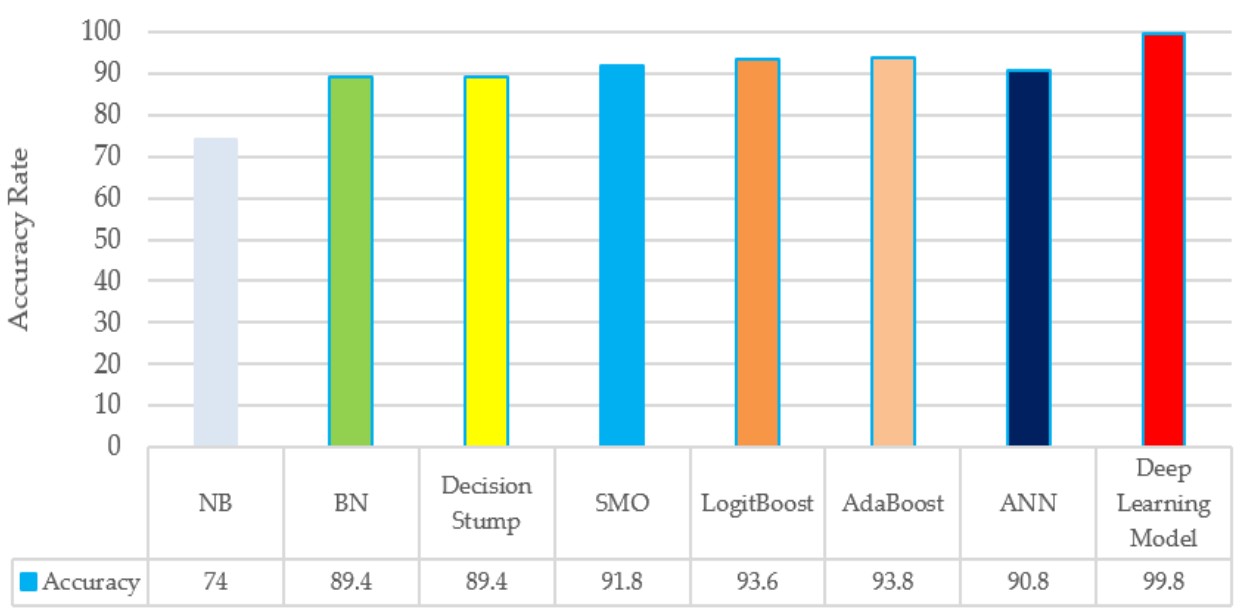

| Accuracy | NB | BN | Decision Stump | SMO | LogitBoost | AdaBoost | ANN | Deep Learning Model |
|---|---|---|---|---|---|---|---|---|
| Accuracy | 74 | 89.4 | 89.4 | 91.8 | 93.6 | 93.8 | 90.8 | 99.8 |

Learning Algorithms

**Figure 4.** Comparing accuracy of our deep learning results with machine learning classifiers.

Tables 5 and 6 indicate the DL performances as well as shallow ANN results on ten thousand malware and benign samples from different malware types and various malicious categories. We employed three dense hidden layers each with 500, 64, and 32 neurons, respectively. To effectively measure the DL model results, cross-validation with different *k* values (*k* = 5 and *k* = 10) and different proportional splits (85%, 15%, 70%, 30%, 50%, 50%) have been used. The performances are evaluated based on the well-known measures, including precision (Prec), recall (Rec), f-measure (F1), and accuracy (Acc) for both benign (0) and malware (1) classes. We achieved 99–100% precision, 99–100% recall, 99.5–100% f-measure, and 99.6–99.8% accuracy when divergent cross-validation parameters and various percentage splits were used (Table 5). These results indicated that we almost classified all the tested samples correctly as malicious and benign from several categories. When the ANN was used for classification, the performances decreased sharply. For instance, the precision measured as 70%, 73%, 77% for benign class; 95%, 99%, 99% for malware class, and the accuracy measured as 86%, 86.7%, 90.8%, which were much lower than the used DL model (Table 6). Similar lower test results were obtained when other metrics were used, including recall and f-measure, with ANN.

Table 7 demonstrates confusion matrices when DL and ANN are used as a classifier for different test cases. It can be seen from Table 7 that 4 out of 1500 and 20 out of 5000 samples were misclassified, which was relatively low when compared with scientific studies results, which shows the efficiency of the proposed model. We could not obtain the same performance when ANN was used as a classifier. Figure 3 indicates the loss function values with respect to the number of epochs when separating malware from benign files. The loss function value measured as 0.2, which was relatively low at the beginning, and declined over time even further to 0.001, almost 0 when epochs was 50.

Table 8 shows the traditional machine learning classifiers' results on collected malicious and benign samples. These classifiers were NB, BN, Decision Stump, SMO, LogitBoost, and AdaBoost. NB results were relatively poor based upon precision, recall, f-measure, and accuracy measures. BN, Decision Stump, and SMO classifiers performed better than NB but still were not satisfactory based on precision, recall, f-measure, and accuracy (Table 8). LogitBoost and AdaBoost classifiers performed the best among the other machine learning classifiers as well as ANN, but they performed poorly when compared with the DL model. The DL model results were substantially high when they were compared with traditional machine learning classifiers. For example, the accuracy rate was measured as 99.8% when DL was used versus 74%, 89.4%, 89.4%, 91.8%, 93.6%, 93.8%, and 90.8% when NB, BN, Decision Stump, SMO, LogitBoost, AdaBoost, and ANN were used, respectively (Figure 4). We obtained such a high performance ratio with the DL model over traditional machine learning algorithms because the DL model identifies high-level features from data level by level in hidden layers. Some features can be observed in both benign and malware samples, which results in misclassification in traditional machine learning algorithms. Those features can be reduced with hidden layers in DL by generating high-level features. In this case, most of the time, DL can discover hidden patterns in data more than ANN and other ML algorithms. When we increased the number of hidden layers, such as four, five, or more, the performance did not increase much. Thus, to keep the model simple as well as obtain optimal solutions, we have selected three hidden layers in which neurons are decreasing in the subsequent layers.

To accurately assess the proposed feature generation method with appropriate DL model performances, the obtained results were compared with well-known machine learning algorithms and various deep learning architecture results in the literature. Most of the studies that were analyzed in the paper used Microsoft BIG 2015, Malimg, Malevis, or their own datasets. Several studies focused on the classification of malware families, while a few studies distinguished malware from benign samples. The obtained experimental results were considerably better than other methods in the literature (Table 9). For instance, the accuracy was measured as 99.8% for the proposed method while 98.6% [26], 97.7% [30], 96.3% [33], 94.5% [34], 90.4% [35], 95% [36], 89.6% [37], 65.4% [38], 87.4% [39] for studies from the literature, respectively (Table 9). Similar performance values were gathered when precision, recall, f-measure, false positive, and false negative measures were used. It can also clearly be observed from Table 8 that, most of the time, deep-learning-based models produce a higher accuracy rate than traditional machine learning methods (NB, SVM, XGBoost, etc.). The distribution of the data as well as the quality of the data have a major effect on DL model performance. Generally, the datasets used for literature studies have redundant features and are not up to date. Further, there are no benign samples in some of these datasets, and there are not enough balanced samples for each malware category. These deficiencies in data reduce the model performances. On the other hand, our dataset is new, has a balance on the distribution of samples (malware category and benign), and is pre-processed well before being provided to the DL model. Additionally, the different hyperparameter settings as well as hidden layers with a different number of neurons increased DL model performances for our test case as well.

**Table 9.** Comparing the proposed method results with other existing literature studies that use deep learning and machine.

| Paper | Year | Dataset | Model | Performance Based on Accuracy (%) |
|---|---|---|---|---|
| Kumar [26] | 2021 | Malimg and Microsoft BIG 2015 | Deep transfer learning | 98.63 |
| Aslan and Yilmaz [30] | 2021 | Malimg, Microsoft BIG 2015, and Malevis | Hybrid deep learning architecture | 97.78 |
| Kim et al. [33] | 2017 | Microsoft BIG 2015 | Transferred generative adversarial network | 96.36 |
| Cui et al. [34] | 2018 | Malimg | Deep learning using CNN | 94.5 |
| Vinayakumar et al. [35] | 2019 | Malimg | Deep neural networks | 90.4 |
| Saxe and Berlin [36] | 2015 | Their own dataset | Deep neural network | 95 |
| Santos et al. [37] | 2013 | Their own dataset | Machine learning using SVM | 89.6 |
| Firdausi et al. [38] | 2010 | Their own dataset | Machine learning using NB | 65.4 |
| Bozkir et al. [39] | 2021 | Their own dataset | Machine learning using XGBoost | 87.45% |
| Proposed Model | 2022 | Our own dataset | Deep learning | 99.80 |

## 5. Limitations and Future Works

Although the proposed deep-learning-based malware detection system can effectively detect several malware samples within the different families, there are a few limitations in the paper that need to be mentioned. In this study, the program samples are detected as malware or benign. Further classification has not been performed to show the types of malware. In the next study, we aim to specify the types of malware samples as well as test our proposed architecture on other malware datasets, including Malimg and Microsoft BIG 2015. The proposed model can detect new malware variants, but we did not test our model with crafted input for adversary attacks. In the future, it will be tested for evasion attacks as well. Moreover, we can analyze more malware and benign samples for future work. The training time of the model was in an acceptable time interval. In this study, we utilized a deep learning method for classification; in the future, we would like to combine various deep learning architectures with reinforcement learning to build a more resistant system. In the future, we aim to use new technologies with deep learning, including blockchain, cloud computing, and big data, to increase the model performances as well as provide more computational power and resources.

## 6. Conclusions

The paper suggested a new malware detection system that consists of three modules: program sample collection, feature extraction, and classification. In the sample collection module, several malwares, as well as benign samples, are collected from different sources and labeled by using VirusTotal and desktop marking. Some basic features that are gathered during the malware analysis process are taken into consideration during the labeling stage. In some cases, it is difficult to label the malware samples by using an antivirus scanner; for those cases, extra features are used. In the feature generation module, malware and benign samples are analyzed under dynamic analysis tools and system calls execution traces are collected. In this module, system calls are divided into five categories including process, file, registry, memory, and network. By using these categories, first behaviors are created, and then features are generated from the behaviors. In this phase, behaviors themselves, the locations in which they are executed, as well as some of the other parameters, including PID, parent PID, results, time, event class, and sequence, are considered. At the end of this module, feature frequency and row feature vector are generated. In the classification module, the novel deep learning methodology is proposed to recognize the different

malware and benign samples. The methodology consists of one input, three hidden, and one output layer. In hidden layers, fully connected 500, 64, and 32 neurons are used in the first, second, and third layers, respectively. To increase the model performance and use more significant features, various activation functions in order of Sigmoid, ReLU, Sigmoid, and Softmax are used.

The experiment test results showed that the proposed model that uses a different deep learning methodology performed well on our created malware dataset. Even though using different hyperparameters and performance metrics, most of the time, the obtained DR and accuracy were measured as 99% and 99.80%, respectively. This shows that combining the appropriate malware analyzing method with an appropriate DL architecture can efficiently separate different malware samples from the benign ones. The proposed model can efficiently detect various malwares, including both traditional as well as new variants from the benign files. When different machine learning algorithms were used for the same dataset, the obtained performance values were decreased. Even ANN performed poorly when compared with the proposed DL methodology. The proposed model has also been compared with the leading methods in the literature. According to the comparison, the proposed model performed better than well-known state-of-the-art studies based on the DR, precision, recall, f-measure, and accuracy metrics.

**Funding:** This research received no external funding.

**Data Availability Statement:** The data used to support the findings of this study are available from the author upon the proper request.

**Conflicts of Interest:** The author declares no conflict of interest.

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
