# Peer review of "Separating Malicious from Benign Software Using Deep Learning Algorithm"

_electronics, doi:10.3390/electronics12081861_

Round 1
Reviewer 1 Report
Grammatical corrections
Line 328 - the word "is" before "evades"
Line 721 - word "is" before "performed" and "it is" before "compared"
Figure 1- The vertical text in each section of the neural network is too large and does not fit into the individual rectangles.
The proposed model uses neural network architecture (3 dense hidden layers each with 500,64 and 32 neurons). It is not stated in the paper for what reason this particular architecture was chosen.
In the introduction of the paper it is mentioned that the advantage of using a deep neural network allows the detection of zero-day vulnerabilities unlike other methods, but in the evaluation of the tests of the proposed method this is not considered in any way, the author only mentions in chapter 5 that they did not test this feature.
It would be interesting to describe what architecture of the automated testing environment was used to learn and test the proposed method.
The author of the paper describes that 10,000 sample programs were used for testing. In the evaluation, the proposed solution shows an impressive success rate compared to other methods. To what extent is this success rate affected by the fact that each method uses a different dataset?
Author Response
Reviewer #1 Concerns:
Comments and Suggestions for Authors:
Concern 1: Grammatical corrections
Line 328 - the word "is" before "evades"
Line 721 - word "is" before "performed" and "it is" before "compared"
Author’s Response
We would like to thank the reviewer for this valuable comment. We have made necessary corrections based on the reviewer request.
Concern 2: Figure 1- The vertical text in each section of the neural network is too large and does not fit into the individual rectangles.
Author’s Response
We would like to thank the reviewer for this valuable comment. Figure 1 vertical text scaled down in order to fit to the individual rectangles.
Concern 3: The proposed model uses neural network architecture (3 dense hidden layers each with 500,64 and 32 neurons). It is not stated in the paper for what reason this particular architecture was chosen.
Author’s Response
We would like to thank the reviewer for this valuable comment. To keep the model simple as well as get optimal solutions, we have selected 3 hidden layers in which neurons are decreasing in the following subsequent layers. We obtained best results when using 3 hidden layers with 500, 64, 32 neurons. Based on the reviewer request, we added appropriate text in the appropriate locations in the text.
Concern 4: In the introduction of the paper it is mentioned that the advantage of using a deep neural network allows the detection of zero-day vulnerabilities unlike other methods, but in the evaluation of the tests of the proposed method this is not considered in any way, the author only mentions in chapter 5 that they did not test this feature.
Author’s Response
We would like to thank the reviewer for this valuable comment. We updated the related sections based on the reviewer request.
Concern 5: It would be interesting to describe what architecture of the automated testing environment was used to learn and test the proposed method.
Author’s Response
We would like to thank the reviewer for this valuable comment. We added some detail of the testing environment. We would like to extend our architecture and share more details in the next study.
Concern 6: The author of the paper describes that 10,000 sample programs were used for testing. In the evaluation, the proposed solution shows an impressive success rate compared to other methods. To what extent is this success rate affected by the fact that each method uses a different dataset?
Author’s Response
We would like to thank the reviewer for this valuable comment. In the comparison table 9, we compared various DL architectures performances on different datasets. Some of the datasets are private which means they haven't been published anywhere else, thus we cannot test our model in those datasets. The other datasets such as Malimg, Microsoft BIG 2015, and Malevis datasets are publicly available and generally used for classification, and require more work for us to perform our model in those datasets. Actually, in 2021 we published one paper which used different DL model on datasets Malimg, Microsoft BIG 2015, and Malevis. We have also tested our model on a simple smaller dataset, we obtained similar results which we didn't put in the paper. For now, we put in the limitation section these datasets and will test on these datasets in the next study as a feature work.

Reviewer 2 Report
The article is devoted to the separation of malware from safe programs using deep learning.
The topic of the article is relevant.
The article has the following shortcomings, which must be addressed before subsequent publication:
1. Present the purpose of the study and the objectives of the points
2. Research methodology needs to be improved. Map out your exploration using the mind mapping tool.
3. Lines 43-56 largely repeat table 1. We must eliminate duplication. What is the meaning of table 1 if there is one column? It is better to make a list or a diagram. Or add a column with numeric data to the table so that it really is a statistic.
4. The numbering of the figures does not match: p. 650 - should be Fig. 4 (instead of Fig. 5), p. 635 - should be Fig. 3 (instead of Fig. 4). Or is there a drawing missing?
5. Fig. 2 - it is not clear what the colors of the circles symbolize. Write explanations. Why were these software units chosen, what variants and families?
6. Paragraph p. 421-435 - Present this description as a flowchart of an algorithm
7. The “conclusions” section needs to be rewritten, it is not suitable in this form. It is necessary to list specifically the items done, in accordance with the tasks of the work
8. Please present the known methods in the form of a diagram (section 2.5)
9. P. 309 - Table II (Table 2 is required), p. 396 - Table III (required - table 3).
10. Lines 501-505 , 526-531 - align the formulas
Author Response
Reviewer #2 Concerns:
Comments and Suggestions for Authors: The article is devoted to the separation of malware from safe programs using deep learning. The topic of the article is relevant, but article has the following shortcomings, which must be addressed before subsequent publication:
Concern 1: Present the purpose of the study and the objectives of the points
Author’s response: We would like to thank the reviewer for this valuable comment. Based on the reviewer request, the purpose of the study is mentioned more clearly along the text.
Concern 2: Research methodology needs to be improved. Map out your exploration using the mind mapping tool.
Author’s response: We would like to thank the reviewer for this valuable comment. Methodology is updated based on the reviewer request.
Concern 3: Lines 43-56 largely repeat table 1. We must eliminate duplication. What is the meaning of table 1 if there is one column? It is better to make a list or a diagram. Or add a column with numeric data to the table so that it really is a statistic.
Author’s response: We would like to thank the reviewer for this valuable comment. The caption of the table is updated and the amount of repetition is declined.
Concern 4: The numbering of the figures does not match: p. 650 - should be Fig. 4 (instead of Fig. 5), p. 635 - should be Fig. 3 (instead of Fig. 4). Or is there a drawing missing?
Author’s response: We would like to thank the reviewer for this valuable comment. The wrong numbering of figures are corrected.
Concern 5: Fig. 2 - it is not clear what the colors of the circles symbolize. Write explanations. Why were these software units chosen, what variants and families?
Author’s response: We would like to thank the reviewer for this valuable comment. The circles are colored based on the reviewer's request. And a short explanation added into the text. The circles are colored based on the malware types in level as well as different malware families.
Concern 6: Paragraph p. 421-435 - Present this description as a flowchart of an algorithm
Author’s response: We would like to thank the reviewer for this valuable comment. We didn't understand what the reviewer meant by this comment.
Concern 7: The “conclusions” section needs to be rewritten, it is not suitable in this form. It is necessary to list specifically the items done, in accordance with the tasks of the work
Author’s response: We would like to thank the reviewer for this valuable comment. The conclusion section is updated based on the reviewer request.
Concern 8: Please present the known methods in the form of a diagram (section 2.5)
Author’s response: We would like to thank the reviewer for this valuable comment. The important DL-based detection systems are summarized in Table 2. In order to avoid repetition, we think that we should not present the same information in different diagrams.
Concern 9: P. 309 - Table II (Table 2 is required), p. 396 - Table III (required - table 3).
Author’s response: We would like to thank the reviewer for this valuable comment. Based on the reviewer request, the Table 2 and table 3 references in the text are updated.
Concern 10: Lines 501-505 , 526-531 - align the formulas
Author’s response: We would like to thank the reviewer for this valuable comment. Based on the reviewer request, the formulas are aligned.

Reviewer 3 Report
In this paper, the authors proposed using deep learning methodology to distinguish malware from benign samples.
Several important issues are required to be addressed to improve the paper quality:
1- In Introduction Section, the authors mentioned that the motivation for using deep learning is the automatic extraction of features. However, the extraction of features in the proposed model was conducted through some steps mentioned in Subsection 3.1. Dataset Creation and Representation Method. Please, it is important to clarify the motivation.
2- In Section 3. Materials and Methods
- In lines 349-350, the authors mentioned “To increase the detection performance, domain expert knowledge and deep learning model are combined in this study”. Please explain this sentence. Where this combination is explained in this section.
- In 3.2.1. Using the Proposed Deep Learning Methodology to Detect Malware
The authors claimed to use deep learning architecture to distinguish malware from benign samples. I think the proposed deep learning architecture is not common deep learning but similar to the traditional ANN with multilayers. Please explain the difference between the proposed deep learning architecture and the traditional ANN with multilayers. Besides, demonstrate using the references that the proposed model is deep learning.
- I suggest removing Section 3.2.2. Using Machine Learning Algorithms to Detect Malware since the other machine learning did not perform well in the results section.
- In line 568, the authors mentioned that 10.000 program samples were used in this study. It is important to explain how many malware samples and benign samples as well.
3- In Section 4. Experimental Results and Discussions
The results need more discussion. The authors mentioned that the proposed overcomes the ANN, other machine learning, and other previous works. However, the authors did not discuss and justify the results.
4- In Section 5. Limitations and Future Works
It is recommended to show the training time of the proposed deep learning methodology to distinguish malware from benign samples. It is expected that using DL with the proposed architecture including three hidden layers 500, 64, and 32 is too slow.
Author Response
Reviewer #3 Concerns:
Comments and Suggestions for Authors: In this paper, the authors proposed using deep learning methodology to distinguish malware from benign samples. Several important issues are required to be addressed to improve the paper quality:
Concern 1- In Introduction Section, the authors mentioned that the motivation for using deep learning is the automatic extraction of features. However, the extraction of features in the proposed model was conducted through some steps mentioned in Subsection 3.1. Dataset Creation and Representation Method. Please, it is important to clarify the motivation.
Author’s response: We would like to thank the reviewer for this valuable comment. Based on the reviewer request, the explanation is added into the related part of the introduction section. Deep learning can be used for various purposes in learning processes including feature extraction, classification, and dimensionality reduction. Besides, it can be combined with other ML models to enhance performance. We assume that even with domain expert knowledge, the DL model may perform better in some cases.
Concern 2- In Section 3. Materials and Methods
- In lines 349-350, the authors mentioned “To increase the detection performance, domain expert knowledge and deep learning model are combined in this study”. Please explain this sentence. Where this combination is explained in this section.
- In 3.2.1. Using the Proposed Deep Learning Methodology to Detect Malware
The authors claimed to use deep learning architecture to distinguish malware from benign samples. I think the proposed deep learning architecture is not common deep learning but similar to the traditional ANN with multilayers. Please explain the difference between the proposed deep learning architecture and the traditional ANN with multilayers. Besides, demonstrate using the references that the proposed model is deep learning.
- I suggest removing Section 3.2.2. Using Machine Learning Algorithms to Detect Malware since the other machine learning did not perform well in the results section.
- In line 568, the authors mentioned that 10.000 program samples were used in this study. It is important to explain how many malware samples and benign samples as well.
Author’s response: We would like to thank the reviewer for this valuable comment. DL can be used for many ways: classification, feature extraction, dimensionality reduction. In this study, we used domain knowledge to extract some distinguishing patterns in collected data to create a dataset. Features can be in any level in this case, we can take each system call as a feature or use several system calls to combine a feature. Or after using several system calls to create each feature, we can perform some other operation to create a higher level feature from the previous layer feature. For instance, for feature extraction, DBN, RBM, CNN, SAE and GAN DL model can be used while for classification DNN, LSTM, RNN and RBM can be used. After we extracted features at some level, we used 3 hidden layers with several neuron DL models with different activation functions to separate malware from the benign ones. In other worlds, the existing features are used in some level of hidden layers neurons to extract the patterns which can be mostly seen in malware while rarely seen in benign files. In our case there is no problem to call it DL, because it is a bit different from the classical MLP. MLP is not really different from the DL, but arguably just one type of DL. Back-propagation theoretically allows you to train a network with many layers. But before the advent of DL, researchers have not had widespread success training neural networks with more than 2 layers. This was generally because of vanishing or exploding gradients. Before the DL, MLPs were initialized using random numbers. But today, MLPs used the gradient of the network's parameters with respect. to the network's error to adjust the parameters to better values in each training iteration. In back propagation, to evaluate this gradient involves the chain rule and you must multiply each layer's parameters and gradients together across all the layers. This is a lot of multiplication, especially for networks with more than 2 layers. If most of the weights across many layers are less than 1 and they are multiplied many times then eventually the gradient just vanishes into a machine-zero and training stops. If most of the parameters across several layers are greater than 1 and they are multiplied many times then eventually the gradient explodes into a huge number and the training process becomes intractable. DL suggested a new initialization strategy: use a series of single layer networks - which does not suffer from vanishing gradients. Besides, we have already used classical ANN in our test case, and the performance was pretty low when it compared with our DL results. We added ‘Section 3.2.2. Using Machine Learning Algorithms to Detect Malware’ to compare the DL results with ML results. Because DL outperformed the ANN and ML algorithms in our study. To make this clear, we would like to keep this subsection. The number of malware and benign samples that are analyzed are added into the text as a reviewer request.
Concern 3- In Section 4. Experimental Results and Discussions
The results need more discussion. The authors mentioned that the proposed overcomes the ANN, other machine learning, and other previous works. However, the authors did not discuss and justify the results.
Author’s response: We would like to thank the reviewer for this valuable comment. Based on the reviewer request, in the ‘Experimental Results and Discussions’ section the reasons of why the proposed model outperformed the state of the art studies as well as traditional machine learning algorithms are explained clearly.
Concern 4- In Section 5. Limitations and Future Works
It is recommended to show the training time of the proposed deep learning methodology to distinguish malware from benign samples. It is expected that using DL with the proposed architecture including three hidden layers 500, 64, and 32 is too slow.
Author’s response: We would like to thank the reviewer for this valuable comment. The training time of DL architecture was not slow, because we performed a pre-processing stage before running the DL algorithm. Preprocessing in the data generation phase made the data suitable for deep learning. In this context, 1 sentence is added into the ‘Limitations and Future Works’ section.
Round 2
Reviewer 2 Report
In this version, the authors have made some changes. However, there are still unresolved issues that should be resolved for a better perception of the article by the reader. First of all, this concerns the need to present some text fragments in the form of figures, diagrams. I think that 4 drawings for such a large article (22 pages) is not enough.
1) Fig. 2 - it is already more clear that malware is grouped into different groups. Please sign what the blue, white, pink, black and yellow circles mean, what groups of malware they symbolize
2) Table 1 contains one column (I already wrote about it). Please add a column numbered (1,2,3). And it is desirable to add another 3rd column, for example, the degree of importance of this trend.
3) Regarding the presentation of your research methodology in the form of a mind map. Here is an example of how you can imagine it.
https://www.mindmeister.com/ru/542554281/academic-mind-map-for-research-papers
You can also present your methodology in the form of a parametric scheme - input data, output, toolkit. This will allow the reader to more clearly see the main point of your research.
4) paragraph c. 421-435 (according to the new line numbering 430-444). Present this description in the form of a flowchart of an algorithm.
Answer: “We would like to thank the reviewer for this valuable comment. We did not understand what the reviewer meant by this comment.”
I would like you to present your verbal description, where there are several conditions, in the form of a flowchart or block diagram. This will make it better for the reader.
Author Response
Reviewer #2 Concerns:
Comments and Suggestions for Authors: In this version, the authors have made some changes. However, there are still unresolved issues that should be resolved for a better perception of the article by the reader. First of all, this concerns the need to present some text fragments in the form of figures, diagrams. I think that 4 drawings for such a large article (22 pages) is not enough.
Author’s Response: Thanks to the reviewer for the concern. We have used 4 figures and 9 tables, a total of 14 visual items to explain the proposed malware detection system. Some of the figures and tables are so long that can only fit into one page. We did not want to overwhelm the readers by putting extra tables or figures that is why we have not put extra figures.
Concern 1: Fig. 2 - it is already more clear that malware is grouped into different groups. Please sign what the blue, white, pink, black and yellow circles mean, what groups of malware they symbolize
Author’s Response: Thank you very much for this valuable concern. Based on the reviewer’s request, the following text is added to the text where the explanation was given for Figure 2. In Figure 2, different colors are used to show the malware types in level as well as various malware families. For instance, yellow colors represent the different types of malware samples while blue colors show the subclass of malware types. In this regard, downloader, dropper, injector, and adware can be seen as a subclass of Trojan while Keyloggers can be seen as a subclass of spyware. Pink colors symbolize new malware variants or more complex malware which use encryption or packed evasion techniques, while black colors represent various malware families.
Concern 2: Table 1 contains one column (I already wrote about it). Please add a column numbered (1,2,3). And it is desirable to add another 3rd column, for example, the degree of importance of this trend.
Author’s Response: Thank the reviewer for this comment. It was a bit difficult, but we added 1 more column to Table 1 with parameter class. Now Table 1 has 2 columns.
Concern 3: Regarding the presentation of your research methodology in the form of a mind map. Here is an example of how you can imagine it. https://www.mindmeister.com/ru/542554281/academic-mind-map-for-research-papers You can also present your methodology in the form of a parametric scheme - input data, output, toolkit. This will allow the reader to more clearly see the main point of your research.
Author’s Response: Thanks to the reviewer for the suggestion. We have given the research methodology in Figure 1 which only can fit into one big page. In Figure 1, we visualized all the procedures applied in the model from the data collection phase to the detection phase step by step. We continued to explain the model in the following paragraphs for each module (three modules). Since Figure 1 is large and can be also divided into 3 different figures, we think that there is no extra figure needed in this section.
Concern 4: paragraph c. 421-435 (according to the new line numbering 430-444). Present this description in the form of a flowchart of an algorithm. Answer: “We would like to thank the reviewer for this valuable comment. We did not understand what the reviewer meant by this comment.” I would like you to present your verbal description, where there are several conditions, in the form of a flowchart or block diagram. This will make it better for the reader.
Author’s Response: Thanks to the reviewer for the suggestion. Between lines 430-444, there are some feature representation examples given for n-grams as well as the suggested method for data creation. Instead of drawing a flowchart or using different figure representations, we decided to mention it in the paragraph to keep it simple and more understandable for the readers.

Reviewer 3 Report
The author addressed most of the comments in a good way
Author Response
Reviewer #3 Concerns: The author addressed most of the comments in a good way
Author’s Response: Thank you very much to the reviewer for the previous suggestions.
